# Bridging the Gap between Supervised and Self-supervised Contrastive Learning

## Abstract

Compared to supervised learning, self-supervised learning has progressed more empirically than theoretically. Many successful algorithms combine multiple techniques that are supported by experiments. While there are some theoretical works, few have explicitly formulated its connection to supervised learning. To address this gap, we take a principled approach. We theoretically formulate a self-supervised learning problem as an approximation of a supervised learning problem in the context of contrastive learning. From the formulated problem, we derive a loss that is closely related to existing contrastive losses, thereby providing a foundation for these losses. The concepts of prototype representation bias and balanced contrastive loss are naturally introduced in the derivation, which provide insights to help understand self-supervised learning. We discuss how components of our framework align with practices of self-supervised learning algorithms, focusing on SimCLR. We also investigate the impact of balancing the attracting force between positive pairs and the repelling force between negative pairs. The proofs of our theorems are provided in the appendix, and the code to reproduce experimental results is provided in the supplementary material.

## 1 Introduction

Representation learning, the process of acquiring condensed but meaningful representations (Bengio et al., 2013; LeCun et al., 2015; Goodfellow et al., 2016), lies at the core of advancing machine learning capabilities. Conventional supervised learning depends heavily on labeled data. It can be problematic in the face of diverse and dynamic real-world data. Human annotation is not scalable due to its labor-intensive requirement and not generalizable due to its subjective nature. Furthermore, it is error-prone (Vasudevan et al., 2022; Beyer et al., 2020; Shankar et al., 2020).

Amidst these challenges, self-supervised learning has emerged as a new paradigm, supported by the notion that humans primarily learn from unlabeled data (Orhan et al., 2020; Savage, 2019). It has demonstrated success in various fields, including but not limited to computer vision, natural language processing, and speech recognition (Ozbulak et al., 2023; Schiappa et al., 2023; Gui et al., 2023).

However, compared to supervised learning, self-supervised learning has been more empirically driven.[1] The mainstream approach is to adopt Siamese networks as base architecture and combine various engineering techniques, such as memory banks, momentum encoders, stop-gradient, projectors, predictors, multi-crop, and centering (Wu et al., 2018; He et al., 2020; Grill et al., 2020; Chen & He, 2021; Caron et al., 2020; 2021; Purushwalkam & Gupta, 2020; Zbontar et al., 2021; Amrani et al., 2022). The techniques are often explained intuitively, and their performance is supported by experiments. This approach may not be satisfactory since it can obscure what problem the algorithms are addressing essentially.

In this paper, we theoretically formulate a self-supervised learning problem and derive its solution. To do so, we observe that self-supervised learning is more nuanced compared to unsupervised learning. It not

---

[1]Self-supervised learning is sometimes metaphorically referred to as the dark matter of intelligence, implying that its principle is not easily understood despite its significant impact (Balestriero et al., 2023).

only utilizes unlabeled data but also generates its own labels from it.[2] This suggests a connection between supervised and self-supervised learning. However, this connection has largely been addressed implicitly through experiments and has not been elucidated properly. Therefore, it is difficult to say we have a satisfactory theory linking supervised and self-supervised learning. To explore this connection, we first cast a supervised learning problem as an optimization problem and then extend to formulate a self-supervised learning problem, leveraging natural approximations. Subsequently, we convert the objective function into a more manageable form under certain assumptions. Then, eventually, our problem is reduced to minimizing an upper bound of the objective function.

Our framework provides an explanation of the problem that self-supervised learning solves. We show that the loss induced from our objective function is closely related to the normalized temperature-scaled cross-entropy (NT-Xent) loss in SimCLR (Chen et al., 2020a), which serves as a hub for many algorithms. We introduce the concept of *prototype representation bias*, which arises naturally during the approximation process. It provides insight into a data augmentation strategy. We also introduce a loss inspired by our framework, which we term the *balanced contrastive loss*. We then emphasize the significance of striking the balance between attracting and repelling components of the loss. As a result, our work helps understand self-supervised learning in a more structured and systematic way.

**Contributions** of our work are summarized as follows:

1. We propose a unified theoretical framework that formalizes self-supervised learning as an approximation of supervised learning, bridging a critical gap in the literature.

2. From the theoretical framework, we derive a mathematical foundation for commonly used contrastive losses, particularly InfoNCE-type losses.

3. The framework unifies common practices and explains the coexistence of asymmetric and symmetric approaches, enhancing understanding of the field.

4. We introduce prototype representation bias and balanced contrastive loss, offering insights into self-supervised learning and the role of balancing parameters.

## 2 Related work

**Contrastive losses**  Our work falls in the category of contrastive learning characterized by contrastive loss. The concept of contrastive loss was introduced in Chopra et al. (2005). From this, several different types of contrastive losses has emerged. The triplet loss simultaneously considers three representations, each serving as an anchor, a positive sample, and a negative sample (Weinberger & Saul, 2009; Chechik et al., 2010). Furthermore, the $(m+1)$-tuplet loss treats $m+1$ representations: an anchor, a positive sample, and $m-1$ negative samples, and it is composed in the form of a softmax function (Sohn, 2016). Wu et al. (2018) combines a temperature parameter and proximal regularization to have the noise-contrastive estimation (NCE) loss. The NT-Xent loss (equivalently, the InfoNCE loss (Oord et al., 2018)) is obtained by constructing a cross-entropy form loss using $2m$ augmented images from a minibatch of $m$ images (Chen et al., 2020a). Wang & Isola (2020) takes the contrastive loss as a given and shows that it asymptotically promotes alignment and uniformity in representations. In Khosla et al. (2020), the concept of contrastive loss is applied in reverse to the supervised setting. In our work, we lay a foundation for the contrastive losses.

**Views on self-supervised learning**  There have been attempts to interpret contrastive learning within different conceptual frameworks. There is an approach that provides unified views bridging contrastive learning and covariance-based learning (Huang et al., 2021; Garrido et al., 2022; Lee et al., 2021; Balestriero & LeCun, 2022; Tian et al., 2020). There is another approach that interprets contrastive learning as maximizing the mutual information of positive pairs (Hjelm et al., 2018; Oord et al., 2018; Bachman et al., 2019; Wang

---

[2]This is implied within expressions such as pseudo labels (Doersch et al., 2015; Noroozi & Favaro, 2016; Zhang et al., 2016; Gidaris et al., 2018), target (or teacher) encoders (Tarvainen & Valpola, 2017; He et al., 2020; Grill et al., 2020; Chen & He, 2021; Caron et al., 2021; Oquab et al., 2023) in the literature.

& Isola, 2020; Li et al., 2021; Aitchison & Ganev, 2024). HaoChen et al. (2021) views self-supervised learning as learning spectral embeddings of an augmentation graph. In addition, there have been attempts to frame self-supervised learning through clustering (Caron et al., 2020), bootstrapping (Grill et al., 2020), semi-supervised learning (Chen et al., 2020b), or knowledge distillation (Caron et al., 2021; Oquab et al., 2023). The idea of supervision is often alluded to in various approaches. In this work, we make this idea explicit from a specific perspective. Starting from a formulation of the problem from first principles, we systematically derive how the mechanism of attracting or repelling pseudo-labels mathematically translates into attracting or repelling other samples.

## 3 Problem formulation

In this section, we first formulate a supervised representation learning problem as an optimization problem, followed by its self-supervised counterpart. Throughout the paper, we use uppercase letters to denote random elements, lowercase letters to denote non-random elements (including realizations of the random elements), and calligraphic letters to denote sets.

### 3.1 Supervised representation learning problem

Let $\mathcal{X} \times \mathcal{Y}$ be a dataset comprising images and their associated visual concepts (represented as labels) of interest. To exploit the dataset to the fullest, we consider a set of transformations $\mathcal{T}$ that preserve the visual concepts and leverage them to create an augmented dataset.[3] Then, we define the augmented dataset induced by $\mathcal{T}$ as

$$\mathcal{T}(\mathcal{X}) \times \mathcal{Y}$$
$$:= \{(t(x), y) : (x, y) \in \mathcal{X} \times \mathcal{Y} \text{ and } t \in \mathcal{T}\}. \quad (1)$$

Equipped with the augmented dataset, we want to train an encoder $f_\theta : \mathcal{X} \to \mathbb{R}^d \setminus \{0\}$ which is parameterized by learnable parameters $\theta$. It maps an image $t(x)$ to its representation $f_\theta(t(x))$. Typically, the representation dimension $d$ is small relative to the image size. By training the encoder, our goal is to make representations of images with the same visual concept, gathered close together, while representations of images with different visual concepts are meaningfully distant from each other. To keep the theoretical framework intuitive and concise, we begin with just these two fundamental ideas: positive samples are clustered, while negative samples are separated.

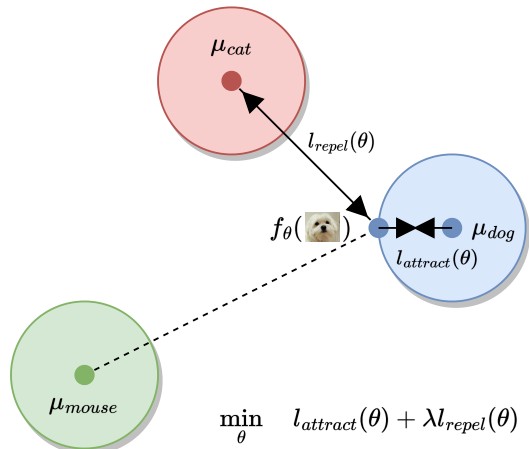

$$\min_\theta \quad l_{attract}(\theta) + \lambda l_{repel}(\theta)$$

Figure 1: **Supervised learning as an optimization.** The loss $l_{\text{attract}}(\theta)$ encourages the image representation to attract the prototype representation $\mu_{\text{dog}}$ that shares the visual concept of that image. On the other hand, the loss $l_{\text{repel}}(\theta)$ prompts the image representation to repel the prototype representation $\mu_{\text{cat}}$ that is closest among those not sharing the visual concept of that image. The parameter $\lambda$ balances the two losses.

To achieve our goal, we employ the concept of *prototype representation* of a visual concept to set targets for images (Li et al., 2020; Caron et al., 2020). This denotes a point in the representation space that embodies the visual concept. To see the whole approximation process, we start by assuming that an oracle gives the ideal prototype representation, which can serve as a common target for images with the same visual concept during training. However, since such an oracle does not exist in reality, we later construct the prototype representation using available data.

---

[3]Note that the choice of data augmentation can also be seen as a type of supervision (Xiao et al., 2020). By treating the labels of augmented images as identical, we supervise the resolution at which the model should be transformation invariant. Therefore, unlike $\mathcal{X}$, $\mathcal{T}(\mathcal{X})$ contains partial information about the labels, which enables self-supervised learning.

From now on, we tag a data point $(t(x), y) \in \mathcal{T}(\mathcal{X}) \times \mathcal{Y}$ and base the formulation on it. Let $l_{\text{attract}}(\theta)$ and $l_{\text{repel}}(\theta)$ denote the attracting and repelling components of the loss function for the image representation $f_\theta(t(x))$. Specifically, $l_{\text{attract}}(\theta)$ encourages similarity with the prototype representation $\mu_y$ of its own label, while $l_{\text{repel}}(\theta)$ penalizes similarity with the prototype representations $\mu_{y'}$ of other labels $(y' \neq y)$. The similarity measure is usually chosen to be cosine similarity. Then, we formulate the supervised representation learning problem as the following optimization problem:

$$\min_\theta \quad l_{\text{attract}}(\theta) + \lambda l_{\text{repel}}(\theta) \tag{2}$$

where $\lambda > 0$ is a parameter which balances the two losses.

In contrastive learning, there is no need to repel negative samples that are already dissimilar enough. In this context, we only repel the prototype representation with the maximum similarity among those representing distinct labels. Then, our problem becomes as follows:

$$\min_\theta \quad -s\left(f_\theta(t(x)), \mu_y\right) + \lambda \max_{y' \neq y} s\left(f_\theta(t(x)), \mu_{y'}\right) \tag{3}$$

where $s(\cdot, \cdot)$ is a similarity measure. For a better understanding, refer to Figure 1.

Note that our formulation is similar to minimizing the triplet loss in spirit (Chechik et al., 2010; Schroff et al., 2015; Schultz & Joachims, 2003; Arora et al., 2019). In our formulation, we can see $f_\theta(t(x))$ as the anchor, the prototype representation $\mu_y$ as the positive sample, and the prototype representation $\mu_{y'}$ as the negative sample. Only considering the negative sample with maximum similarity is related to the concept of hard negative mining (Girshick, 2015; Faghri et al., 2017; Oh Song et al., 2016). This idea has sometimes been implemented through the introduction of the concept of support vectors or margin (Cortes & Vapnik, 1995; Schroff et al., 2015). Pursuing this to the extreme leads us to repel the most challenging example, namely, the negative sample with maximum similarity.

Now, we construct the prototype representations. For a given label $y$, a natural choice for the prototype representation of the label is the expectation of the representations of the images with the same label, i.e.,

$$\hat{\mu}_y := \mathbb{E}_{T,X|y} f_\theta(T(X)) \tag{4}$$

where $T$ is distributed over $\mathcal{T}$, and $X$ is conditionally distributed over $\{x : (x, y) \in \mathcal{X} \times \mathcal{Y}\}$. Plugging it to Equation (3), our problem becomes as follows:

$$\min_\theta \quad -s\left(f_\theta(t(x)), \mathbb{E}_{T,X|y} f_\theta(T(X))\right) + \lambda \max_{y' \neq y} s\left(f_\theta(t(x)), \mathbb{E}_{T',X'|y'} f_\theta(T'(X'))\right) \tag{5}$$

where $T'$ and $X'$ are independent copies of $T$ and $X$, respectively.

### 3.2 Self-supervised representation learning problem

In the self-supervised learning regime, we do not have access to the labels. So, we use a surrogate prototype representation for the image $t(x)$ as the target. We construct it as the expectation of the representations of the *available* images sharing the same label as $t(x)$, i.e.,

$$\tilde{\mu}_y := \mathbb{E}_T f_\theta(T(x)). \tag{6}$$

In Section 5, we demonstrate the importance of finding a data augmentation strategy that approximates well from the prototype representation $\mathbb{E}_{T,X|y} f_\theta(T(X))$ to the surrogate prototype representation $\mathbb{E}_T f_\theta(T(x))$. Plugging it in the attracting component of Equation (5), we rewrite our problem as follows:

$$\min_\theta \quad -s\left(f_\theta(t(x)), \tilde{\mu}_y\right) + \lambda \max_{y' \neq y} s\left(f_\theta(t(x)), \hat{\mu}_{y'}\right). \tag{7}$$

Note that we leave the repelling component as is since it can be managed without modification. In Section 4, we find an upper bound of the above objective function, and in Section 5, we show the upper bound can be minimized using a Siamese network. Through this, we show how attracting and repelling pseudo-labels ($\tilde{\mu}_y$ and $\hat{\mu}_{y'}$) can be achieved through attracting and repelling samples ($f_\theta(t'(x))$ and $f_\theta(t'(x'))$). Refer to Figure 2 for a better understanding.

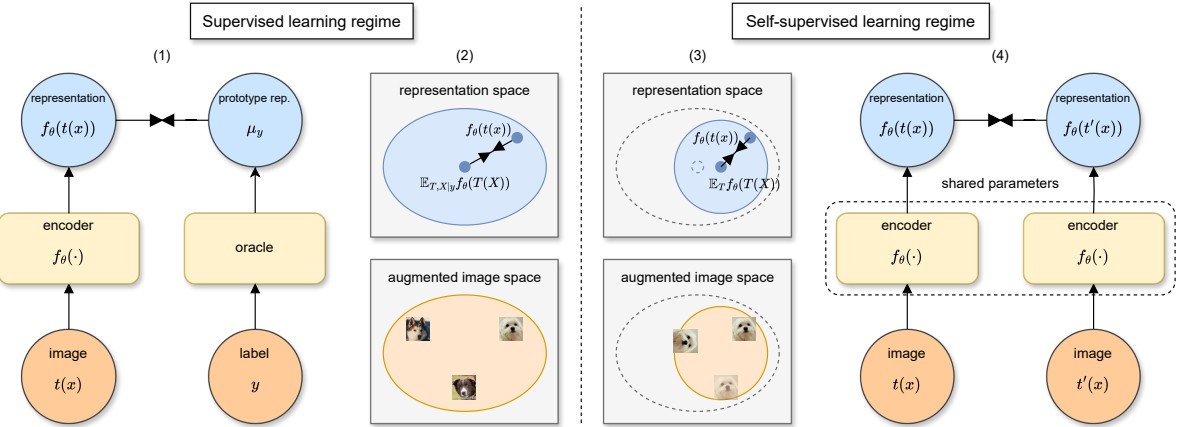

Figure 2: **Self-supervised learning as an approximation of supervised learning.** (1) In an ideal supervised regime, the ideal prototype representation $\mu_y$ is given by an oracle. (2) In a realistic supervised regime, the prototype representation is constructed as the expectation $\mathbb{E}_{T,X|y}f_\theta(T(X))$ of the representations of the images with the same label $y$. (3) In a self-supervised regime, a surrogate prototype representation is constructed as the expectation $\mathbb{E}_T f_\theta(T(x))$ of the representations of the available images sharing the same label as $t(x)$. (4) This can be effectively implemented using a Siamese network.

## 4  Theoretical derivation

In this section, we determine upper bounds of the attracting and repelling components. Our objective is to minimize these upper bounds, addressing the optimization problem discussed in the previous section. We show that the *triplet loss with pseudo-labels* can be interpreted as an approximation to an *InfoNCE-type loss with samples*. This perspective provides a theoretical link between prototype-based supervised learning and contrastive self-supervised learning frameworks.

### 4.1  Attracting component

We first find an upper bound for the attracting component by making the following assumptions based on common practice.

**Assumption 4.1** (cosine similarity)**.** The similarity measure $s(\cdot,\cdot)$ is cosine similarity, i.e., $s(x_1, x_2) = x_1 \cdot x_2/(\|x_1\|\|x_2\|)$. When we say $s(x_1, x_2)$, we assume $x_1$ and $x_2$ are nonzero.

**Assumption 4.2** ($l_2$-normalization)**.** Representations at the end of the encoder are $l_2$-normalized so that $\|f_\theta(t(x))\| = 1$, i.e., $f_\theta : \mathcal{X} \to \mathbb{S}^{d-1}$. Here, $\mathbb{S}^{d-1} := \{x \in \mathbb{R}^d : \|x\| = 1\}$ denotes the unit sphere in $\mathbb{R}^d$.

**Assumption 4.3** (technical assumption)**.** We additionally make a technical assumption which means that the two vectors $f_\theta(t(x))$ and $\mathbb{E}_T f_\theta(T(x))$ lie in the same hemisphere, i.e., $f_\theta(t(x)) \cdot \mathbb{E}_T f_\theta(T(x)) \geq 0$. Informally speaking, this means that the augmentation does not distort the image too much, so $\mathbb{E}_T f_\theta(T(x))$ does not point in a completely different direction.

**Theorem 4.4** (upper bound of the attracting component)**.** *Assume Assumption 4.1, 4.2, and 4.3 hold. Then,*

$$- s\left(f_\theta(t(x)), \mathbb{E}_T f_\theta(T(x))\right) \leq -\mathbb{E}_T s\left(f_\theta(t(x)), f_\theta(T(x))\right). \tag{8}$$

*Proof.* Refer to Appendix A.1.1. □

We approximate the upper bound and obtain the following sample analog:

$$\widetilde{l}_{\text{attract}}(\theta) := -\frac{1}{|\hat{\mathcal{T}}|} \sum_{t' \in \hat{\mathcal{T}}} s\left(f_\theta(t(x)), f_\theta(t'(x))\right) \tag{9}$$

where $\hat{\mathcal{T}}$ is the set of transformation samples.

## 4.2 Repelling component

We now find an upper bound for the repelling component by making the following assumption.

**Assumption 4.5** (balanced dataset). Labels are uniformly distributed, i.e., $p(y) = \frac{1}{n}$, where $n$ is the finite number of labels.

**Theorem 4.6** (upper bound of the repelling component). *Assume Assumption 4.1, 4.2, and 4.5 hold. Let $\nu := \min_{y' \neq y} \|\mathbb{E}_{T', X'|y'} f_\theta(T'(X'))\|$. Then, for all $\alpha > 0$,*

$$\max_{y' \neq y} s\left(f_\theta(t(x)), \mathbb{E}_{T', X'|y'} f_\theta(T'(X'))\right) \leq \mathbb{E}_{T'}\left[\frac{1}{\nu\alpha} \log \mathbb{E}_{X'} \exp\left(\alpha s\left(f_\theta(t(x)), f_\theta(T'(X'))\right)\right)\right] + \frac{1}{\nu\alpha} \log n. \quad (10)$$

*Proof.* We approximate the maximum function by the log-sum-exp function and apply Jensen inequality to pull out the expectations. For the detailed proof, refer to Appendix A.1.2. $\square$

If we approximate the upper bound and trim the constant terms, which are not relevant to optimization, we obtain the following:

$$\widetilde{l}_{\text{repel}}(\theta) := \frac{1}{|\hat{\mathcal{T}}|} \sum_{t' \in \hat{\mathcal{T}}} \frac{1}{\nu\alpha} \log \sum_{x' \in \hat{\mathcal{X}}} \exp(\alpha s(f_\theta(t(x)), f_\theta(t'(x')))) \quad (11)$$

where $\hat{\mathcal{T}}$ is the set of transformation samples, and $\hat{\mathcal{X}}$ is the set of image samples.

## 4.3 Total loss

By combining Equation (9) and (11), the total loss $\widetilde{l}(\theta) := \widetilde{l}_{\text{attract}}(\theta) + \lambda \widetilde{l}_{\text{repel}}(\theta)$ is as follows:

$$\widetilde{l}(\theta) = \frac{1}{|\hat{\mathcal{T}}|} \sum_{t' \in \hat{\mathcal{T}}} \left[-s\left(f_\theta(t(x)), f_\theta(t'(x))\right) + \frac{\lambda}{\nu}\left[\frac{1}{\alpha} \log \sum_{x' \in \hat{\mathcal{X}}} \exp(\alpha s(f_\theta(t(x)), f_\theta(t'(x'))))\right]\right]. \quad (12)$$

By rearranging, we have

$$\widetilde{l}(\theta) = \frac{1}{\alpha|\hat{\mathcal{T}}|} \sum_{t' \in \hat{\mathcal{T}}} \left[-\log \frac{\exp(\alpha s\left(f_\theta(t(x)), f_\theta(t'(x))\right))}{\left(\sum_{x' \in \hat{\mathcal{X}}} \exp(\alpha s(f_\theta(t(x)), f_\theta(t'(x'))))\right)^{\lambda/\nu}}\right]. \quad (13)$$

Note that this equation and the NT-Xent in SimCLR are similar in their forms, which we discuss in more detail in the next section.

# 5 Understanding self-supervised learning

In this section, we discuss components of self-supervised learning algorithms within our framework, focusing on SimCLR (Chen et al., 2020a), which has served as a central point for many algorithms.

In experiments, we utilize SimCLR with a temperature parameter $\tau$ of 0.5, employing ImageNet (Deng et al., 2009) as the dataset and ResNet-50 (He et al., 2016) as the backbone. We assess top-1 accuracy using linear evaluation, a standard protocol for evaluating self-supervised learning algorithms. Note that, since ImageNet contains 1,000 classes, the chance-level accuracy is 0.1%. For a fair comparison, all settings are kept the same except for the specific factor under investigation. For the detailed implementation, refer to A.3.

### 5.1 Architecture: Siamese networks

When approximating the upper bound $-\mathbb{E}_T s\left(f_\theta(t(x)), f_\theta(T(x))\right)$ in Equation (8), we compare the similarity between two representations $f_\theta(t(x))$ and $f_\theta(t'(x))$. This is suitable for implementation by a Siamese network (Bromley et al., 1993). We augment a single image $x$ to obtain two differently augmented $t(x)$ and $t'(x)$. Then, we pass them through the two encoders $f_\theta$ that share the parameter $\theta$ and compare the similarity of the outputs. So, our derivation shows that considering similarity with the prototype representation aligns well with using a Siamese network, which is a common architecture in self-supervised learning.

Siamese networks are fundamentally symmetric in that the two encoders often have the same architecture and share parameters. However, there are algorithms aimed at enhancing performance by introducing asymmetry into Siamese networks (He et al., 2020; Chen & He, 2021; Grill et al., 2020; Caron et al., 2020; 2021; Oquab et al., 2023; Tian et al., 2021). In such cases, it is empirically shown to be helpful to ensure that the variance of the outputs from one encoder is lower than that from the other encoder (Wang et al., 2022). The encoder with the lower variance is referred to as the target or teacher encoder, and the encoder with the higher variance is referred to as the source or student encoder. In our problem formulation, the original attracting component in Equation (8) is $-s\left(f_\theta(t(x)), \mathbb{E}_T f_\theta(T(x))\right)$ where the two attracting objects $f_\theta(t(x))$ and $\mathbb{E}_T f_\theta(T(x))$ are asymmetric. Note that $\mathbb{E}_T f_\theta(T(x))$ can be approximated by $\frac{1}{n}\sum_{i=1}^{n} f_\theta(T_i(x))$, and $\frac{1}{n}\sum_{i=1}^{n} f_\theta(T_i(x))$ has less variance than $f_\theta(T(x))$. So, our problem formulation and Theorem 4.4 may provide insight to understand why there exist both symmetry and asymmetry themes in the self-supervised learning literature.

### 5.2 Loss: NT-Xent

Let $\{x_1, \ldots, x_m\}$ be a minibatch of $m$ images. If we transform each image in two different ways and pass them through the encoder, we obtain representation pairs $\{(f_\theta(t(x_i)), f_\theta(t'(x_i))) : i = 1, \ldots, m\}$ of $2m$ augmented images, which we denote as $\{(z_i, z_i') : i = 1, \ldots, m\}$. Then, in the case of $\lambda = \nu$, the summand in Equation (13) can be implemented as

$$-\log \frac{\exp(\alpha s(z_i, z_i'))}{\sum_{j \in [m]\setminus\{i\}} \exp(\alpha s(z_i, z_j'))} \tag{14}$$

where $[m] := \{1, \ldots, m\}$.

On the other hand, in the NT-Xent loss used in SimCLR, if we let the temperature parameter $\tau$ be $1/\alpha$, the NT-Xent loss is represented as

$$-\log \frac{\exp(\alpha s(z_i, z_i'))}{\sum_{j \in [m]} \exp(\alpha s(z_i, z_j')) + \sum_{j \in [m]\setminus\{i\}} \exp(\alpha s(z_i, z_j))}. \tag{15}$$

This is a variant of Equation (14). Having the second summation in the denominator can be seen as a method to fully exploit the provided representations, since $(z_i, z_j)$ are also negative pairs when $j \neq i$. When considering the first summation in the denominator, Yeh et al. (2022) empirically demonstrated that it performs better when the sum is over $[m]\setminus\{i\}$ as in Equation (14) rather than $[m]$. Expressions such as cross-entropy and temperature frame contrastive losses in the form of the Boltzmann (or Gibbs) distribution. Our framework offers another perspective on the losses.

### 5.3 Data augmentation: debiased prototype representation

When transitioning from supervised to self-supervised learning, we approximate the prototype representation $\mathbb{E}_{T,X|y} f_\theta(T(X))$ with the surrogate prototype representation $\mathbb{E}_T f_\theta(T(x))$. Therefore, we investigate whether accuracy increases as the two become closer. For this purpose, we define the *prototype representation bias* as

$$\text{Bias}_{\text{proto}} := \mathbb{E}_{(X_0, Y_0)} \|\mathbb{E}_{T,X|Y_0} f_\theta(T(X)) - \mathbb{E}_T f_\theta(T(X_0))\|. \tag{16}$$

We then compare the values by changing the distribution of $T$ through data augmentation. We compare SimCLR's default data augmentation (`base`) with cases where we exclude Gaussian blur (`-gaussian_blur`) and color distortion (`-color_distortion`), and with cases where we include random cutout (`+random_cutout`) and random rotation (`+random_rotation`), resulting in a total of five scenarios.

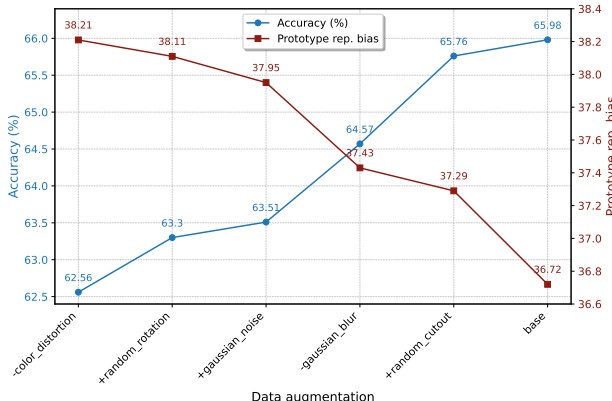

Figure 3: **Accuracy vs. prototype representation bias.** We investigate the relationship between accuracy and prototype representation bias by adding or removing transformations from SimCLR's data augmentation strategy (base). Lower prototype representation bias tends to result in higher accuracy.

Figure 3 shows that using data augmentation with debiased prototype representation leads to an increase in accuracy. It also shows the default data augmentation of SimCLR achieves the highest accuracy while exhibiting the smallest prototype representation bias. Despite the expectation that enriching data augmentation by adding transformations would be beneficial for training, the accuracy still decreases. This may be because the added transformations exacerbate the prototype representation bias.

### 5.4 Similarity measure: cosine similarity with normalized representations

When computing similarity between two representations, many self-supervised learning algorithms including SimCLR normalize the representations and calculate cosine similarity as in Assumption 4.1 and 4.2. To empirically show the significance of these assumptions, we compare three cases: 1) cosine similarity with normalization, 2) dot product without normalization, and 3) negative Euclidean distance without normalization.[4] Table 1 shows that normalization is crucial. Without normalization, the accuracy in the case of negative Euclidean distance is higher than that of dot product. This may be because Euclidean distance measures spatial dissimilarity in a more straightforward manner.

Table 1: Comparison of similarity measures with and without $l_2$-normalization. The results show that cosine similarity with normalization significantly outperforms the other variants.

| Similarity measure | | |
| --- | --- | --- |
| CS w/ $l_2$ | Dot w/o $l_2$ | -Eucl. w/o $l_2$ |
| 65.98 | 0.43 | 10.63 |

### 5.5 Dataset: balanced

Previous studies have shown that contrastive learning algorithms tend to perform better on balanced datasets, where labels are uniformly distributed (Assran et al., 2022b;a; Zhou et al., 2022) as in Assumption 4.5. To further investigate this phenomenon, we provide empirical results within our framework, demonstrating its implications in our specific setting. Table 2 displays SimCLR performs better on a balanced dataset compared to an imbalanced one. In

Table 2: Comparison of class distributions. The results show that the uniform class distribution leads to better performance.

| Class distribution | |
| --- | --- |
| Uniform | Long-tailed |
| 20.82 | 13.65 |

---

[4]Note that when dealing with two normalized vectors, cosine similarity is equivalent to the dot product. Additionally, negative Euclidean distance with normalization is equivalent to cosine similarity with normalization since $-\|a - b\|^2 = -2 + 2a \cdot b$.

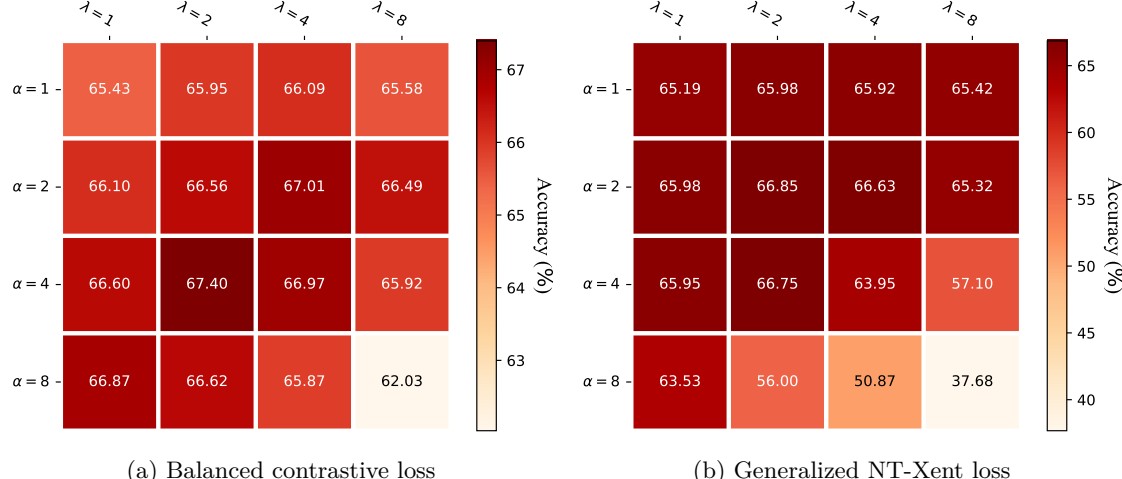

(a) Balanced contrastive loss
(b) Generalized NT-Xent loss

Figure 4: **Impact of balancing parameters $\alpha$ and $\lambda$.** Better balancing can be accomplished through the adjustments of the balancing parameters.

both cases, the training sets contain the same number of images (115,846, which is 9% of the ImageNet training set), but they differ in class distribution. We use an identical test set for both cases. The performance gap illustrates how deviations from this assumption can degrade effectiveness in practice.

## 6 Empirical study

In this section, we introduce a loss that is inspired from the form of Equation (12). To reduce clutter, we rewrite $\lambda/\nu$ as $\lambda$. For one representation $z$ in $2m$ representations generated from a minibatch of $m$ images, we define our loss for the representation as follows:

$$- s(z, z^+) + \lambda \left[ \frac{1}{\alpha} \log \sum_{z^-} \exp(\alpha s(z, z^-)) \right] \tag{17}$$

where $(z, z^+)$ is the positive pair and $(z, z^-)$ are $2(m-1)$ negative pairs. The cost for the whole minibatch is then calculated by taking the mean of the losses of all representations. Note that the attracting component consists of one attracting force, and the repelling component consists of multiple repelling forces. We term the loss the *balanced contrastive loss.*

There are two hyperparameters $\alpha > 0$ and $\lambda > 0$ in the balanced contrastive loss. We refer to these as the *balancing parameters* since each is in charge of two different types of balancing in contrastive learning. The parameter $\alpha > 0$ adjusts the relative magnitudes within the repelling forces (Kalantidis et al., 2020; Zhang et al., 2022; Jiang et al., 2024). Note that the repelling component is a smooth approximation to the maximum function (refer to Lemma A.1 and Wang & Liu (2021)):

$$\lim_{\alpha \to \infty} \left[ \frac{1}{\alpha} \log \sum_{z^-} \exp(\alpha s(z, z^-)) \right] = \max_{z^-} s(z, z^-). \tag{18}$$

If $\alpha$ is large, the repelling forces with representations having high similarities contribute more in the overall repelling component. In self-supervised learning, negative samples may have images with the same label (called sampling bias in Chuang et al. (2020)). So, if we make $\alpha$ too large, there is a risk of repelling images with the same label. Therefore, setting the value of $\alpha$ appropriately can be thought of as hedging the risk with multiple negative samples. This also offers insight into the role of the temperature parameters of InfoNCE-type losses. On the other hand, the parameter $\lambda > 0$ adjusts the relative magnitudes of the attracting and repelling forces.

To study the impact of balancing parameters $\alpha$ and $\lambda$, we test our loss over a grid of parameters $\{(\alpha, \lambda) : \alpha, \lambda \in \{1, 2, 4, 8\}\}$. We also investigate the case where the positive pair is included in the summation in Equation (17). We call the case the generalized NT-Xent loss here since it is equivalent to the NT-Xent loss when $\lambda = 1$. Figure 4 illustrates the changes in accuracy based on various combinations of the parameters. The balanced contrastive loss generally achieves higher maximum accuracies than the generalized NT-Xent loss in this experiment.

For the balanced contrastive loss, the highest accuracy is achieved when $(\alpha, \lambda) = (4, 2)$, and for the generalized NT-Xent loss, the highest accuracy is achieved when $(\alpha, \lambda) = (2, 2)$. In both cases, the highest accuracy is not achieved when $\lambda = 1$. This highlights the significance of the balancing parameter $\lambda$. Additionally in both scenarios, it is crucial for $\alpha$ to have an appropriate value that is not too large or too small. Specifically for the generalized NT-Xent, it is advantageous to set $\alpha$ to a smaller value compared to the balanced contrastive loss. This may be due to the presence of the positive sample in the repelling component, meaning that increasing $\alpha$ results in a larger repulsion of the positive sample. Given that the chance-level accuracy for ImageNet is 0.1, this performance difference is notable, achieved solely through weight adjustments. The results also suggest that the current forms of contrastive losses may be limited.

## 7 Discussion

The potential connection between supervised and self-supervised learning has been implied in practical algorithms. It has been interpreted from perspectives such as bootstrapping, clustering, and knowledge distillation, which can align with our framework. From the bootstrapping perspective (Grill et al., 2020), the idea is to construct targets solely based on the representations without any external input. This aligns with the framework, where prototype representations are built solely from the representations themselves and used as pseudo-labels. In this line of work, a predictor is often employed, which can be seen as an additional module designed to match the pseudo-labels (Chen & He, 2021). From the clustering perspective (Tian et al., 2017; Caron et al., 2020), the goal is to ensure consistency in the cluster assignments of transformed images. This aligns with the framework in that the representations of transformed images converge toward a single prototype representation, guiding them to belong to the same cluster. From the knowledge distillation perspective (Xu et al., 2020; Caron et al., 2021), self-supervised learning involves a teacher network transferring knowledge to a student network. This aligns with the framework in that the output of one encoder serves as a prototype representation, guiding the output of the other encoder to match it in the formula we ultimately want to optimize.

## 8 Conclusion

In this work, a self-supervised representation learning problem is theoretically conceptualized as an approximation of a supervised representation learning problem. We first formulate the supervised learning problem concisely and then investigate how its natural approximation arises in the absence of labels. We break down the process into individual steps, allowing the community to focus on improving each step. Our framework enhances an understanding of existing algorithms. The loss derived at the end is related to widely used InfoNCE-type losses. Additionally, our framework provides insights into the biases of prototype representations and balancing in contrastive loss, which can be considered when designing an optimal algorithm. It also provides richer context for components of existing algorithms, such as data augmentation, temperature hyperparameters, and symmetric/asymmetric architecture. Our work aims to contribute to building a firm foundation for self-supervised learning. We hope that our work will benefit the self-supervised learning community by serving as a basis and providing guidance for research.

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

# A    Appendix

## A.1    Proofs

This subsection presents the proofs of Theorem 4.4 and Theorem 4.6.

### A.1.1    Proof of Theorem 4.4

We restate the assumptions and the theorem and provide the proof below.

**Assumption 4.1** (cosine similarity)**.** The similarity measure $s(\cdot, \cdot)$ is cosine similarity, i.e., $s(x_1, x_2) = x_1 \cdot x_2/(\|x_1\|\|x_2\|)$. When we say $s(x_1, x_2)$, we assume $x_1$ and $x_2$ are nonzero.

**Assumption 4.2** ($l_2$-normalization)**.** Representations at the end of the encoder are $l_2$-normalized so that $\|f_\theta(t(x))\| = 1$, i.e., $f_\theta : \mathcal{X} \to \mathbb{S}^{d-1}$. Here, $\mathbb{S}^{d-1} := \{x \in \mathbb{R}^d : \|x\| = 1\}$ denotes the unit sphere in $\mathbb{R}^d$.

**Assumption 4.3** (technical assumption)**.** We additionally make a technical assumption which means that the two vectors $f_\theta(t(x))$ and $\mathbb{E}_T f_\theta(T(x))$ lie in the same hemisphere, i.e., $f_\theta(t(x)) \cdot \mathbb{E}_T f_\theta(T(x)) \geq 0$. Informally speaking, this means that the augmentation does not distort the image too much, so $\mathbb{E}_T f_\theta(T(x))$ does not point in a completely different direction.

**Theorem 4.4** (upper bound of the attracting component)**.** *Assume Assumption 4.1, 4.2, and 4.3 hold. Then,*

$$- s\left(f_\theta(t(x)), \mathbb{E}_T f_\theta(T(x))\right) \leq -\mathbb{E}_T s\left(f_\theta(t(x)), f_\theta(T(x))\right). \tag{8}$$

*Proof.*

$$-s\left(f_\theta(t(x)), \mathbb{E}_T f_\theta(T(x))\right) \overset{(i)}{=} -\frac{f_\theta(t(x)) \cdot \mathbb{E}_T f_\theta(T(x))}{\|f_\theta(t(x))\|\|\mathbb{E}_T f_\theta(T(x))\|} \tag{19}$$

$$\overset{(ii)}{=} -\frac{f_\theta(t(x)) \cdot \mathbb{E}_T f_\theta(T(x))}{\|\mathbb{E}_T f_\theta(T(x))\|} \tag{20}$$

$$\overset{(iii)}{\leq} -\frac{f_\theta(t(x)) \cdot \mathbb{E}_T f_\theta(T(x))}{\mathbb{E}_T \|f_\theta(T(x))\|} \tag{21}$$

$$\overset{(iv)}{=} -f_\theta(t(x)) \cdot \mathbb{E}_T f_\theta(T(x)) \tag{22}$$

$$\overset{(v)}{=} -\mathbb{E}_T \left[f_\theta(t(x)) \cdot f_\theta(T(x))\right] \tag{23}$$

$$\overset{(vi)}{=} -\mathbb{E}_T \left[\frac{f_\theta(t(x)) \cdot f_\theta(T(x))}{\|f_\theta(t(x))\|\|f_\theta(T(x))\|}\right] \tag{24}$$

$$\overset{(vii)}{=} -\mathbb{E}_T s\left(f_\theta(t(x)), f_\theta(T(x))\right) \tag{25}$$

where $(i)$ and $(vii)$ are by Assumption 4.1, $(ii)$, $(iv)$, and $(vi)$ are by Assumption 4.2, $(iii)$ is by Assumption 4.3, the convexity of $l^2$-norm (Boyd & Vandenberghe, 2004), and Jensen's inequality, and $(v)$ is by the linearity of expectation. This completes the proof of Theorem 4.4. $\qquad\square$

### A.1.2    Proof of Theorem 4.6

Before we prove Theorem 4.6, we need three additional lemmas. While the proofs of the lemmas are straightforward, they are not readily available in the existing literature. Therefore, we provide them here for the sake of self-containedness.

**Lemma A.1.** *For $\alpha > 0$ and $x_i \in \mathbb{R}$, $i = 1, 2, \ldots, n$,*

$$\max_{i=1,\ldots,n} x_i \leq (1/\alpha) \log \sum_{i=1}^{n} \exp(\alpha x_i) \leq \max_{i=1,\ldots,n} x_i + \frac{\log n}{\alpha}, \tag{26}$$

*where the equalities hold when $\alpha$ goes to infinity.*

*Proof.* We have

$$\exp\left(\max_{i=1,\ldots,n}(\alpha x_i)\right) \le \sum_{i=1}^{n} \exp(\alpha x_i) \le n \exp\left(\max_{i=1,\ldots,n}(\alpha x_i)\right). \tag{27}$$

Since $\alpha > 0$,

$$\alpha \max_{i=1,\ldots,n} x_i \le \log \sum_{i=1}^{n} \exp(\alpha x_i) \le \alpha \max_{i=1,\ldots,n} x_i + \log n. \tag{28}$$

This completes the proof of Lemma A.1. $\square$

**Lemma A.2.** *For $\alpha > 0$ and $x_i \in \mathbb{R}$, $i = 1, 2, \ldots, n$,*

$$u(x_1, \ldots, x_n) := (1/\alpha) \log \sum_{i=1}^{n} \exp(\alpha x_i) \tag{29}$$

*is convex on $\mathbb{R}^n$.*

*Proof.* Note that the log-sum-exp function $v(x_1, \ldots, x_n) := \log \sum_{i=1}^{n} \exp(x_i)$ is convex on $\mathbb{R}^n$ (Boyd & Vandenberghe, 2004; Ghaoui, 2014). $u(x_1, \ldots, x_n) = (1/\alpha)v(\alpha(x_1, \ldots, x_n))$, and composition with an affine mapping preserves convexity (Boyd & Vandenberghe, 2004). Thus, $u(x_1, \ldots, x_n)$ is also convex on $\mathbb{R}^n$. This completes the proof of Lemma A.2. $\square$

**Lemma A.3.** *If $g_1(x) \ge 0$ for all $x$, and $g_2(x) \ge 0$ for some $x$, then*

$$\max[g_1(x)g_2(x)] \le \max[g_1(x)] \max[g_2(x)]. \tag{30}$$

*Proof.* By default, $g_2(x) \le \max[g_2(x)]$. Since $g_1(x) \ge 0$ for all $x$, $g_1(x)g_2(x) \le g_1(x) \max[g_2(x)]$. Taking the maximum of both sides, we have $\max[g_1(x)g_2(x)] \le \max[g_1(x) \max[g_2(x)]]$. Since $g_2(x) \ge 0$ for some $x$, $\max[g_2(x)] \ge 0$, and thus $\max[g_1(x)g_2(x)] \le \max[g_1(x)] \max[g_2(x)]$. This completes the proof of Lemma A.3. $\square$

Now, we are ready to prove Theorem 4.6. We restate the assumption and the theorem and provide the proof below.

**Assumption 4.5** (balanced dataset)**.** Labels are uniformly distributed, i.e., $p(y) = \frac{1}{n}$, where $n$ is the finite number of labels.

**Theorem 4.6** (upper bound of the repelling component)**.** *Assume Assumption 4.1, 4.2, and 4.5 hold. Let $\nu := \min_{y' \ne y} \|\mathbb{E}_{T', X'|y'} f_\theta(T'(X'))\|$. Then, for all $\alpha > 0$,*

$$\max_{y' \ne y} s\left(f_\theta(t(x)), \mathbb{E}_{T', X'|y'} f_\theta(T'(X'))\right) \le \mathbb{E}_{T'}\left[\frac{1}{\nu\alpha} \log \mathbb{E}_{X'} \exp\left(\alpha s\left(f_\theta(t(x)), f_\theta(T'(X'))\right)\right)\right] + \frac{1}{\nu\alpha} \log n. \tag{10}$$

*Proof.*

$$\max_{y' \ne y} s\left(f_\theta(t(x)), \mathbb{E}_{T', X'|y'} f_\theta(T'(X'))\right) \overset{(i)}{=} \max_{y' \ne y} \frac{f_\theta(t(x)) \cdot \mathbb{E}_{T', X'|y'} f_\theta(T'(X'))}{\|f_\theta(t(x))\| \|\mathbb{E}_{T', X'|y'} f_\theta(T'(X'))\|} \tag{31}$$

$$\overset{(ii)}{=} \max_{y' \ne y} \frac{f_\theta(t(x)) \cdot \mathbb{E}_{T', X'|y'} f_\theta(T'(X'))}{\|\mathbb{E}_{T', X'|y'} f_\theta(T'(X'))\|} \tag{32}$$

$$\overset{(iii)}{\le} \frac{1}{\nu} \max_{y' \ne y} \mathbb{E}_{T', X'|y'} s\left(f_\theta(t(x)), f_\theta(T'(X'))\right) \tag{33}$$

where $(i)$ is by Assumption 4.1, $(ii)$ is by Assumption 4.2, and $(iii)$ is by the following argument.

Let $y^*$ be the label that achieves the maximum in Equation (32). Note that under Assumption 4.2, $0 < \|\mathbb{E}_{T', X'|y'} f_\theta(T'(X'))\| \le 1$. If in an ideal case, $f_\theta(t'(x'))$ produces the same representation for every $t'(x')$ that shares the same label $y'$, then $\|\mathbb{E}_{T', X'|y'} f_\theta(T'(X'))\| = \|f_\theta(t'(x'))\| = 1$. To show $(iii)$, we proceed by considering the following two cases.

Case 1: If $f_\theta(t(x)) \cdot \mathbb{E}_{T',X'|y*} f_\theta(T'(X')) \leq 0$, then

$$\frac{f_\theta(t(x)) \cdot \mathbb{E}_{T',X'|y*} f_\theta(T'(X'))}{\|\mathbb{E}_{T',X'|y*} f_\theta(T'(X'))\|} \overset{(i)}{\leq} \frac{f_\theta(t(x)) \cdot \mathbb{E}_{T',X'|y*} f_\theta(T'(X'))}{\mathbb{E}_{T',X'|y*}\|f_\theta(T'(X'))\|} \tag{34}$$

$$\overset{(ii)}{=} f_\theta(t(x)) \cdot \mathbb{E}_{T',X'|y*} f_\theta(T'(X')) \tag{35}$$

$$\overset{(iii)}{=} \mathbb{E}_{T',X'|y*} s(f_\theta(t(x)), f_\theta(T'(X'))) \tag{36}$$

$$\leq \max_{y' \neq y} \mathbb{E}_{T',X'|y'} s(f_\theta(t(x)), f_\theta(T'(X'))) \tag{37}$$

$$\overset{(iv)}{\leq} \frac{1}{\nu} \max_{y' \neq y} \mathbb{E}_{T',X'|y'} s(f_\theta(t(x)), f_\theta(T'(X'))) \tag{38}$$

where $(i)$ is by Jensen's inequality, $(ii)$ is by Assumption 4.2, $(iii)$ is by a similar argument in the proof of Theorem 4.4, and $(iv)$ follows from the fact that $0 < \nu \leq 1$.

Case 2: If $f_\theta(t(x)) \cdot \mathbb{E}_{T',X'|y*} f_\theta(T'(X')) > 0$, then

$$\frac{f_\theta(t(x)) \cdot \mathbb{E}_{T',X'|y*} f_\theta(T'(X'))}{\|\mathbb{E}_{T',X'|y*} f_\theta(T'(X'))\|} \overset{(i)}{\leq} \max_{y' \neq y} \frac{1}{\|\mathbb{E}_{T',X'|y'} f_\theta(T'(X'))\|} \max_{y' \neq y} \left[ f_\theta(t(x)) \cdot \mathbb{E}_{T',X'|y'} f_\theta(T'(X')) \right] \tag{39}$$

$$= \frac{1}{\nu} \max_{y' \neq y} \left[ f_\theta(t(x)) \cdot \mathbb{E}_{T',X'|y'} f_\theta(T'(X')) \right] \tag{40}$$

$$\overset{(ii)}{=} \frac{1}{\nu} \max_{y' \neq y} \mathbb{E}_{T',X'|y'} s(f_\theta(t(x)), f_\theta(T'(X'))) \tag{41}$$

where $(i)$ is by Lemma A.3, and $(ii)$ is by a similar argument in the proof of Theorem 4.4.

Now for brevity, let $g(T'(X')) := s\left(f_\theta(t(x)), f_\theta(T'(X'))\right)$. Then,

$$\max_{y' \neq y} \mathbb{E}_{T',X'|y'} g(T'(X')) \overset{(i)}{\leq} \frac{1}{\alpha} \log \sum_{y' \neq y} \exp\left(\alpha \mathbb{E}_{T',X'|y'} g(T'(X'))\right) \tag{42}$$

$$\overset{(ii)}{\leq} \frac{1}{\alpha} \log \sum_{y'} \exp\left(\alpha \mathbb{E}_{T',X'|y'} g(T'(X'))\right) \tag{43}$$

$$= \frac{1}{\alpha} \log \sum_{y'} \exp\left(\alpha \mathbb{E}_{T'} \mathbb{E}_{X'|y'} g(T'(X'))\right) \tag{44}$$

$$\overset{(iii)}{\leq} \mathbb{E}_{T'} \left[ \frac{1}{\alpha} \log \sum_{y'} \exp\left(\alpha \mathbb{E}_{X'|y'} g(T'(X'))\right) \right] \tag{45}$$

$$\overset{(iv)}{\leq} \mathbb{E}_{T'} \left[ \frac{1}{\alpha} \log \sum_{y'} \mathbb{E}_{X'|y'} \exp\left(\alpha g(T'(X'))\right) \right] \tag{46}$$

$$\overset{(v)}{=} \mathbb{E}_{T'} \left[ \frac{1}{\alpha} \log \left( n \sum_{y'} p(y') \mathbb{E}_{X'|y'} \exp\left(\alpha g(T'(X'))\right) \right) \right] \tag{47}$$

$$= \mathbb{E}_{T'} \left[ \frac{1}{\alpha} \log \left( n \mathbb{E}_{Y'} \mathbb{E}_{X'|Y'} \exp\left(\alpha g(T'(X'))\right) \right) \right] \tag{48}$$

$$= \mathbb{E}_{T'} \left[ \frac{1}{\alpha} \log \left( n \mathbb{E}_{X'} \exp\left(\alpha g(T'(X'))\right) \right) \right] \tag{49}$$

$$= \mathbb{E}_{T'} \left[ \frac{1}{\alpha} \log \left( \mathbb{E}_{X'} \exp\left(\alpha g(T'(X'))\right) \right) \right] + \frac{1}{\alpha} \log n. \tag{50}$$

where $(i)$ is by Lemma A.1, $(ii)$ is by the positivity of $\exp(\alpha x)$ and the monotonicity of $\log(x)$, $(iii)$ is by Lemma A.2 and Jensen's inequality, $(iv)$ is by the convexity of $\exp(\alpha x)$, Jensen's inequality, and the monotonicity of $\log(x)$, and $(v)$ is by Assumption 4.5. This completes the proof of Theorem 4.6. $\qquad\square$

## A.2 Cross-reference

Table 3 shows how each component of SimCLR corresponds to specific parts of our problem formulation and theoretical derivation.

Table 3: Cross-reference between SimCLR and our framework. We compare the key components and provide references to the corresponding sections and theorems.

| Component | SimCLR | Our framework |
|---|---|---|
| Architecture | Siamese network | Subsection 4.1 and 4.2 |
| Loss | NT-Xent | Subsection 4.3 |
| Data augmentation | debiased prototype representation | Subsection 3.2 |
| Similarity measure | cosine similarity with normalization | Theorem 4.4 and 4.6 |
| Dataset | balanced | Theorem 4.6 |

## A.3 Implementation details

This subsection offers a comprehensive description of the implementation details for our experiments. Readers can also refer to the code provided in the supplementary material. With 8 NVIDIA V100 GPUs, the pretraining takes about 2.5 days and 13 GB peak memory usage, the linear evaluation takes about 1.5 days and 8 GB peak memory usage, and the $k$-nearest neighbors takes about 40 minutes and 30 GB peak memory usage.

### A.3.1 Base setting

**Dataset** We use ImageNet as the benchmark dataset, as it is one of the most representative large-scale image datasets. The training set comprises 1,281,167 images, while the validation set comprises 50,000 images. As ImageNet's test set labels are unavailable, we utilize the validation set as a test set for evaluation purposes. ImageNet encompasses 1,000 classes.

**Data augmentation** The following data transformations are sequentially applied during pretraining. Due to variations in image sizes, they are first cropped to dimensions of $224 \times 224$.

- `RandomResizedCrop`: Randomly crop a patch of the image within the scale range of $(0.2, 1)$, then resize it to dimensions of $(224, 224)$.

- `ColorJitter`: Change the image's brightness, contrast, saturation, and hue with strengths of $(0.4, 0.4, 0.4, 0.1)$ with a probability of 0.8.

- `RandomGrayscale`: Convert the image to grayscale with a probability of 0.2.

- `GaussianBlur`: Apply the Gaussian blur filter to the image with a radius sampled uniformly from the range $[0.1, 2]$ with a probability of 0.5.

- `RandomHorizontalFlip`: Horizontally flip the image with a probability of 0.5.

- `Normalize`: Normalize the image using a mean of $(0.485, 0.456, 0.406)$ and a standard deviation of $(0.229, 0.224, 0.225)$.

**Network architecture** The encoder consists of a backbone followed by a projector. We employ ResNet-50 as the backbone and a three-layered fully-connected MLP as the projector. For the projector, the input and output dimensions of all layers are set to 2,048. Batch normalization (Ioffe & Szegedy, 2015) is applied to all layers, and the ReLU activation function is applied to the first two layers.

**Pretraining configuration** We pretrain the encoder with a batch size of 512 for 100 epochs. We employ the SGD optimizer and set the momentum to 0.9, the learning rate to 0.1, and the weight decay rate to 0.0001. Additionally, we implement a cosine decay schedule for the learning rate, as proposed by Loshchilov & Hutter (2016); Chen et al. (2020a).

**Evaluation configuration**  After pretraining, we employ linear evaluation, which is the standard evaluation protocol. We take and freeze the pretrained backbone and attach a linear classifier on top. The linear classifier is then trained on the training set and evaluated on the test set. Training the linear classifier is conducted with a batch size of 4,096 for 90 epochs, utilizing the LARS optimizer (You et al., 2017).

### A.3.2   Implementation details for Section 5.3

To estimate the value of the prototype representation bias, for each $(x_i, y_i)$ in the ImageNet training set $\mathcal{D}$, we sample $t_i$ from $T$ and $x_i'$ from $X|y_i$ and calculate the deviation $\|f_\theta(t_i(x_i')) - f_\theta(t_i(x_i))\|$. Then, we take the average over the entire $\mathcal{D}$ as follows:

$$\frac{1}{|\mathcal{D}|} \sum_{(x_i, y_i) \in \mathcal{D}} \|f_\theta(t_i(x_i')) - f_\theta(t_i(x_i))\|. \tag{51}$$

So, we consider total 1,281,167 samples, which is equivalent to the number of images in the ImageNet training set.

### A.3.3   Implementation details for Section 5.4

When normalization is not carried out, there is a risk of loss overflow, so we resort to using the log-sum-exp trick. It does not alter the values themselves.

### A.3.4   Implementation details for Section 5.5

We use ImageNet-LT (ImageNet Long-Tailed) as a benchmark for imbalanced datasets. ImageNet-LT is a representative dataset specifically designed to address the challenges associated with imbalanced datasets. It is subsampled across the 1,000 classes of ImageNet, following a Pareto distribution with a shape parameter $\alpha$ of 6. The training set consists of 115,846 images, which is approximately 9% of the entire ImageNet training set. The class with the most images contains 1,280 images, while the class with the fewest has only 5 images. The test set is balanced, consisting of 50,000 images, with each class having exactly 50 images.

We construct ImageNet-Uni (ImageNet Uniform) as a subset of ImageNet to enable a fair comparison. We uniformly sample 115,846 images from the ImageNet training set, matching the size of the ImageNet-LT training set. The test set is configured to be identical to that of ImageNet-LT.

## A.4   Further experiments

In this subsection, we provide additional experimental results. We include results on CIFAR-10 (Krizhevsky et al., 2009). Note that, since CIFAR-10 contains 10 classes, the chance-level accuracy is 10%.

### A.4.1   Implementation details for CIFAR-10 experiments

**Dataset**  The training set comprises 60,000 images, while the test set comprises 10,000 images. CIFAR-10 contains 10 classes, with all images standardized to a fixed size of $32 \times 32$.

**Data augmentation**  The following data transformations are sequentially applied during pretraining.

- `RandomResizedCrop`: Randomly crop a patch of the image within the scale range of $(0.08, 1)$, then resize it to dimensions of $(32, 32)$.

- `RandomHorizontalFlip`: Horizontally flip the image with a probability of 0.5.

- `ColorJitter`: Change the image's brightness, contrast, saturation, and hue with strengths of $(0.4, 0.4, 0.4, 0.1)$ with a probability of 0.8.

- `RandomGrayscale`: Convert the image to grayscale with a probability of 0.2.

- `Normalize`: Normalize the image using a mean of $(0.485, 0.456, 0.406)$ and a standard deviation of $(0.229, 0.224, 0.225)$.

Table 4: Standard evaluations. We report top-1 accuracy on CIFAR-10 and ImageNet using two standard evaluation protocols: $k$-nearest neighbor and linear evaluation. Each result is presented as the mean $\pm$ standard deviation over 5 runs.

| Dataset | Protocol | |
|---|---|---|
| | $k$-NN | Linear eval. |
| CIFAR-10 | $80.32 \pm 0.32$ | $86.08 \pm 0.07$ |
| ImageNet | $51.00 \pm 0.22$ | $67.40 \pm 0.07$ |

Table 5: Comparison of class distributions under balanced contrastive loss. The results show that the uniform class distribution leads to better performance.

| Class distribution | |
|---|---|
| Uniform | Long-tailed |
| 21.24 | 15.01 |

**Network architecture** The encoder consists of a backbone followed by a projector. We employ a variant of ResNet-18 for CIFAR-10 as the backbone and a two-layered fully-connected MLP as the projector. For the projector, the input and output dimensions of the first layer and 512 and 2,048, respectively, and the input and output dimensions of the second layer are 2,048. Batch normalization is applied to all layers, and the ReLU activation function is applied to the first layer.

**Pretraining configuration** We pretrain the encoder with 512 batch size for 200 epochs. We employ the SGD optimizer and set the momentum to 0.9, the learning rate to 0.1, and the weight decay rate to 0.0001.

**Evaluation configuration** Training the linear classifier is conducted with a batch size of 256 over 90 epochs, utilizing the SGD optimizer. We set the momentum to 0.9 and learning rate to 30 and use a cosine decay schedule.

### A.4.2 Standard evaluations

Table 4 presents a set of standard evaluations. Error bars, represented as the mean $\pm$ standard deviation, are reported based on five independent runs. We choose $(\alpha, \lambda)$ as $(4, 2)$ and $(2, 4)$ for ImageNet and CIFAR-10, respectively. We also include $k$-nearest neighbors evaluation. Specifically, we retrieve the $k$ nearest training image representations for a given test image representation. Their respective labels are aggregated using a majority voting process to predict the label for the test image. In ImageNet experiments, $k$ is set to 200, whereas in CIFAR-10 experiments, $k$ is set to 1.

### A.4.3 Impact of balancing parameters on CIFAR-10

As in Section 6, Figure 5 shows that, balancing between the attracting component and the repelling component is important using balancing parameters $\alpha$ and $\lambda$.

### A.4.4 Impact of data imbalance on the balanced contrastive loss

As an extension of Section 5.5, we investigate the impact of data imbalance on the balanced contrastive loss in Table 5. We adopt the balancing parameters $\alpha = 2$ and $\lambda = 1$ for comparison, as the SimCLR loss is equivalent to the generalized NT-Xent loss under this setting. Compared to SimCLR, the balanced contrastive loss exhibits relatively improved performance. Nevertheless, similar to SimCLR, performance is higher when the class distribution is balanced. This observation aligns well with our theoretical framework, which assumes uniformity in class distribution.

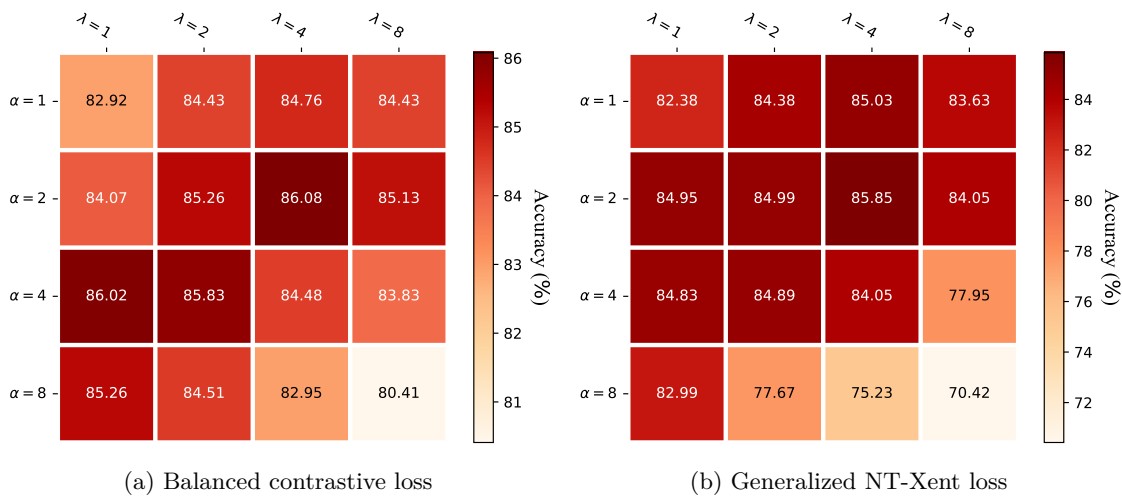

(a) Balanced contrastive loss        (b) Generalized NT-Xent loss

Figure 5: **Impact of balancing parameters $\alpha$ and $\lambda$ on CIFAR-10.** Better balancing can be accomplished through the adjustments of the balancing parameters.

