# OpenReview forum: "Bridging the Gap between Supervised and Self-supervised Contrastive Learning"
_TMLR — Rejected by TMLR_

### Review · Reviewer_CWC2 · 2025-03-28

**Summary Of Contributions:**

This paper formulates a self-supervised learning problem as an approximation of a supervised learning problem, under the assumption that the labels of examples (or the centering of examples belong the same labels)  are given.  It then connect the  derived loss to other self-supervised method,  and ablate the components of the proposed framework with experiments.

**Audience:**

Yes

**Claims And Evidence:**

No

**Requested Changes:**

1. Provide detail comparison to previous work, and clearly show the contributions.

2. Revise the description for avoiding over-claims.

**Strengths And Weaknesses:**

## Strength:
This paper is overall well written. It provides well connection to the supervised learning and self-supervised learning, even though I cannot well recognized its contributions, due to the exists of many previous papers.



## Weaknesses:
1. The overall idea of this paper is assuming the label of examples exists, then the self-supervised learning can be viewed as an approximation of a supervised learning problem. This idea is very similar to the cluster based self-supervised method, e.g, [1] (missing reference), SwAV, and the many papers after it (e..g, DiNO) . In the cluster based method the center of cluster is learned (during training), while this paper is assumed.  Besides, there are many paper provides connection to analyze how different self-supervised learning method (symmetric or asymmetric) can be transfered to a predict problem, e.g. the analyses in Section 5 of SimSiam paper, and [2] (missing reference).  I cannot well recognize the contribution of this paper and its theoretical claims, due to the exists of the previous paper.  I think this paper provide should provide detail comparison to these papers and clearly show the contributions.



2. Some descriptions are not rigorous/ over-claimed (since I cannot well recognize the theoretical contributions):
(1)This paper describe that "unlike supervised learning, self-supervised learning has primarily been driven empirically, with limited emphasis on theoretical foundation.", I believe there are many paper involves the theory of self-supervised learning (see the list of paper [3] [4] [5] [6] [7]).
(2) This paper describe that " Previous works provide valuable insights by either revealing specific aspects or bridging different methods. However, while many approaches allude to the idea of supervision, they do not provide an explanation for how attracting or repelling pseudo-labels mathematically translates into attracting or repelling other samples."  This paper should provide further details to compare the proposed theory to other papers.
(3)This paper describe that "Our work aims to contribute to building a firm foundation for self-supervised learning. ", I cannot well recognize how the paper is "a firm foundation".

3. This paper only provides the analyses on the self-supervised methods, but not propose a new method (therefore no empirical contributions).



Ref:

[1] Self-labelling via simultaneous clustering and representation learning. ICLR 2020.

[2] Exploring the Equivalence of Siamese Self-Supervised Learning via a Unified Gradient Framework. CVPR 2022

[3] A Theoretical Analysis of Contrastive Unsupervised Representation Learning. ICML 2019

[4]Understanding Self-supervised Learning with Dual Deep Networks. ICML 2021.

[5]Understanding Self-Supervised Learning Dynamics without Contrastive Pair. ICML 2021

[6]Provable Guarantees for Self-Supervised Deep Learning with Spectral Contrastive Loss. NeurIPS 2021

[7]Predicting What You Already Know Helps Provable Self-Supervised Learning. NeurIPS 2021

---

> ### Author Response · Authors · 2025-04-20
> **Response to Reviewer CWC2's Comments**
>
> We sincerely thank the reviewer for the thoughtful and constructive feedback. Following your comments, we have carefully revised the manuscript to address the points you raised. We believe that these revisions have strengthened our paper.
>
> ---
> ### **[W1] Cluster-based methods**
>
> We would like to clarify that our contribution is to establish a theoretical framework that connects a prototype-based problem to sample-based contrastive losses.
> Specifically, our work starts from a supervised loss with prototypes, transitions to a self-supervised counterpart, leads to widely used contrastive losses with samples.
>
> Since cluster-based methods leverage prototypes (cluster centroids) for supervision, they are related to our work. So, we refer to these methods throughout the paper (see Section 7 for example).
> We view this relevance not as a weakness, but rather as a strength, since we aim to provide theoretical insight into existing empirical methods.
>
> Regarding the works mentioned:
> - SwAV and DINO are intuitively motivated and empirically validated methods.
> - SeLa [1] tries to maximize the mutual information between positive pairs. This line of work is described in the second paragraph of Section 2.
>
> Rather than analyzing specific methods, our primary focus is to reveal the connection between prototype-based losses and sample-based losses.
> Based on your feedback, we have revised the manuscript to include [1].
>
> ---
> ### **[W2] Existing analysis**
>
> We would like to point out that SimSiam does not offer a formal theoretical analysis.
> While it hypothesizes that self-supervised learning can be interpreted as an expectation-maximization process, this hypothesis was later challenged in [a].
>
> In addition, [2] presents a gradient-based perspective on self-supervised learning, which represents a different viewpoint from ours.
> It is based on the observation that the gradients of different self-supervised learning losses share similar structures, without providing formal proofs.
>
> Unlike these works, we start from a formulated problem. We then state and prove theorems that show how contrastive losses naturally emerge in this setting.
> Based on your feedback, we have revised the manuscript to include [2].
>
> [a] How Does SimSiam Avoid Collapse Without Negative Samples? A Unified Understanding with Self-supervised Contrastive Learning, ICLR 2022
>
> ---
> ### **[W3] On some descriptions**
>
> We thank the reviewer for highlighting statements that may benefit from further clarification. We address each statement below and have revised the manuscript accordingly.
>
> > "unlike supervised learning, self-supervised learning has primarily been driven empirically, with limited emphasis on theoretical foundation."
>
> While there are some theoretical works, our intention was to highlight that, compared to supervised learning, self-supervised learning has been driven more by empirical advances than by theoretical analysis.
>
> We describe theoretical works in Section 2. The differences between the previously uncited works [4,6,7] and our work are described below.
>
> [4] falls under the category of bridging contrastive learning and covariance-based learning as discussed in Section 2.
>
> [6] performs spectral decomposition on a graph constructed from unlabeled samples, in contrast to our supervision-driven approach.
>
> [7] considers an early self-supervised learning approach based on reconstruction pretext tasks. In contrast, our work focuses on recent approaches based on similarity learning or metric learning.
>
>
> > "they do not provide an explanation for how attracting or repelling pseudo-labels mathematically translates into attracting or repelling other samples."
>
> We describe other theoretical works in the preceding text. They do not explicitly derive a connection from a loss with pseudo-labels to a loss with samples. Our contribution lies in bridging this gap by starting from a formulated supervised problem and deriving a principled approximation. That is, we interpret self-supervised learning as a proxy optimization of supervised learning.
>
>
> > "Our work aims to contribute to building a firm foundation for self-supervised learning."
>
> We would like to clarify that our statement was carefully phrased to indicate that we are contributing to an ongoing effort, rather than claiming to provide a complete foundation. That is why we chose the word contribute.
>
>
> ---
> ### **[W4] Empirical contribution**
>
> The balanced contrastive loss derived from our formulation can be seen as a generalization of widely used contrastive losses. Using this loss, we conduct extensive experiments and demonstrate that properly balancing the attracting and repelling components leads to improved performance (see Figure 4).
>
> The accuracy of the standard NT-Xent loss (a special case of the generalized NT-Xent loss with
> $\lambda = 1$) is 65.98%, while the accuracy of the balanced contrastive loss is 67.40%, showing a gap of 1.42% (note that the chance-level accuracy for ImageNet is 0.1%).

---

### Review · Reviewer_4TK3 · 2025-03-29

**Summary Of Contributions:**

This paper presents a theoretical framework that formulates self-supervised learning as an approximation of supervised learning. Authors derive a loss function similar to contrastive losses and introduce concepts like prototype representation bias and balanced contrastive loss to provide insights into self-supervised learning. The paper also discusses how the framework aligns with practices of SimCLR, and investigates the impact of balancing the attracting and repelling forces in the loss.

**Audience:**

Yes

**Broader Impact Concerns:**

I do not identify any immediate broader impact concerns. The work is primarily theoretical and contributes to a better understanding of self-supervised learning. However, as with any representation learning technique, the potential for misuse in downstream applications should be considered (e.g., biased representations leading to unfair outcomes). A brief statement acknowledging this could be added.

**Claims And Evidence:**

Yes

**Requested Changes:**

- Perform an experiment that shows the impact of imbalance in the dataset on the proposed contrastive loss. How does it compare to generalized NT-Xent loss?

- Would it be possible to perform the data augmentation experiment (depicted in Figure 3) for different paths of augmentation (de-)selections?

- Mathematical notation/writing:
	- Formally introduce $\ell_{attract}$ and $\ell_{repel}$ in Equation (2), e.g., by mentioning that those are usually chosen as cosine similarity measure. The second sentence on that page tries to do this but is in my opinion too vague. Also consider using `\text` for attract and repel.
	- Caption in Figure 1: two sentence start with variables which I consider bad style.
	- Some "equations" are missing a LHS/RHS. Consider introducing additional variables for those. Especially, the natural choice for the prototype representation might be worthy of a variable name.
	- Assumption 4.2: Please, formally introduce $\mathbb{S}^{(d-1)}$ as the sphere in an additional clause.
	- Merge the preceding text into Assumption 4.3.
	- The title of Section 5.4 is "cosine similarity with normalization" which is a pleonasm as cosine similarity always assumes that the inputs are normalized. I suggest going with something like "impact of normalization" or comparable.

- Improve captions of Table 1 and 2. Also, the captions of tables and figures in appendix need improvement as they are somewhat vague.

- Very minor comment regarding typography: the title of the manuscript is in title case and the section titles are not. I recommend choosing one style and maintaining consistency.

**Strengths And Weaknesses:**

Strengths:

- The paper provides a principled theoretical formulation of self-supervised learning, offering a deeper understanding of its connection to supervised learning.
- Authors derive a loss function that is under mild assumptions more robust to the choice of the balancing parameters.
- The paper is well-written and the notation is, most of the time, clear. I found it easy to follow and enjoyed reading it.
- Code is provided.


Weaknesses:

- While the connection to SimCLR is discussed, the empirical validation of the proposed balanced contrastive loss and the impact of prototype representation bias could be more extensive.  See requested changes.
- The paper could benefit from a more detailed discussion of the limitations of the proposed framework. For example what happens if not all of the assumptions are met? How does the proposed loss perform on an imbalanced dataset?
- Mathematical notation can be slightly improved at some points. See requested changes.

Overall, this is solid work that I recommend to accept after minor revision.

---

> ### Author Response · Authors · 2025-04-20
> **Response to Reviewer 4TK3's Comments**
>
> We sincerely appreciate your valuable time and effort in reviewing our manuscript. We are especially grateful for your positive evaluation and thorough comments. We believe they have improved the quality of our work. Below, we provide detailed responses to each of your comments, and we have revised the manuscript accordingly.
>
> ---
> ### **[R1] Impact of data imbalance on the balanced contrastive loss**
>
> We perform the requested experiment and present the result in Section A.4.4. As shown, the results align well with our theoretical framework.
>
>
>
> ---
> ### **[R2] Data augmentation**
>
> In our current analysis, we use the standard data augmentation strategy as the base. We then perform ablation studies by adding or removing one transformation at a time. This design choice is to maintain interpretability and align with established practices.
>
> In response to the comment, we have included an additional experiment incorporating the Gaussian noise transformation in the revised manuscript. We believe that the set of transformations considered covers the transformations commonly used in practice.
>
> ---
> ### **[R3] Introducing $l_{\text{attract}}$ and $l_{\text{repel}}$**
>
> To introduce $l_{\text{attract}}$ and $l_{\text{repel}}$ more formally, we have revised the corresponding part as follows.
>
> > Let $l_{\text{attract}}(\theta)$ and $l_{\text{repel}}(\theta)$ denote the attracting and repelling components of the loss function for the image representation $f_{\theta}(t(x))$. Specifically, $l_{\text{attract}}(\theta)$ encourages similarity with the prototype representation $\mu_y$ of its own label, while $l_{\text{repel}}(\theta)$ penalizes similarity with the prototype representations $\mu_{y'}$ of other labels ($y' \neq y$). The similarity measure is usually chosen to be cosine similarity.
>
> ---
> ### **[R4] Caption in Figure 1**
>
> To avoid starting sentences with equations, we have revised the caption as follows.
>
> > The loss $l_{\text{attract}}(\theta)$ encourages the image representation to attract the prototype representation $\mu_{\text{dog}}$ that shares the visual concept of that image. On the other hand, the loss $l_{\text{repel}}(\theta)$ prompts the image representation to repel the prototype representation $\mu_{\text{cat}}$ that is closest among those not sharing the visual concept of that image.
>
>
> ---
> ### **[R5] Introducing variable names**
>
> We have assigned the following variable names to key expressions to improve clarity.
>
> > $\\hat{\\mu}\_{y} := \\mathbb{E}\_{T, X \\vert y}f\_{\\theta}(T(X))$
>
> > $\\tilde{\\mu}\_{y} := \\mathbb{E}\_{T}f\_{\\theta}(T(x))$
>
> > $\\mathrm{Bias}\_{\\text{proto}} := \\mathbb{E}\_{(X\_0,Y\_0)} \\lVert \\mathbb{E}\_{T,X \\vert Y\_0}f\_{\\theta}(T(X)) - \\mathbb{E}\_{T}f\_{\\theta}(T(X\_0)) \\rVert$
>
> ---
> ### **[R6] Assumption 4.2**
>
> We have added the following sentence to Assumption 4.2 to formally introduce $\mathbb{S}^{d-1}$.
>
> > Here, $\mathbb{S}^{d-1} := \\{ x \in \mathbb{R}^d : \|x\| = 1 \\}$ denotes the unit sphere in $\mathbb{R}^d.$
>
> ---
> ### **[R7] Assumption 4.3**
>
> We have merged the preceding text into Assumption 4.3.
>
> ---
> ### **[R8] The title of Section 5.4**
>
> In this context, normalization refers to normalized representations, i.e., $\| f_{\theta}(t(x)) \| = 1$. For example, Equation 20 cannot be derived from Equation 19 using cosine similarity alone. It requires that $f_{\theta}(t(x))$ is already normalized by $f(\cdot)$. For clarity, we have revised the title of Section 5.4 to: “Similarity measure: cosine similarity with normalized representations.”
>
> ---
> ### **[R9] Captions of tables and figures**
>
> We have improved the captions of tables and figures.
>
> ---
> ### **[R10] Title of the manuscript and the section titles**
>
> We have followed the formatting instructions in [TMLR LaTeX stylefile and template](https://github.com/JmlrOrg/tmlr-style-file/archive/refs/heads/main.zip). According to the instructions, the manuscript title is in Title Case, while section titles are in sentence case.

---

### Review · Reviewer_eQZ2 · 2025-04-12

**Summary Of Contributions:**

The paper proposed a theoretical formulation of InfoNCE as an approximation of a supervised triplet loss-based contrastive learning loss.
The proposed formulation starts with a supervised assumption of labels for the input data distribution provided by oracle to compute l_{attract} and l_{repel}. The prototype representation is computed given labels using Eq. 4. For the hard-negative sample the equation resembles that of triple loss given by Eq.3 and 5.

For the Self-supervised learning (SSL) case, since oracle labels are not present, the surrogate prototypes are created as the expectation over samples having the same label, Eq. 6.

Section 4.1 looks into the derivation of the upper bound for the attraction force, while section 4.2 looks into that of the opposing force. This is followed by an analysis of Siamese architecture, data augmentation, similarity measure, l2 normalization, and dataset imbalance discussions.

**Audience:**

No

**Claims And Evidence:**

No

**Requested Changes:**

Please refer to weakness section above.

Additional suggestion:

1. The empirical analysis can be done in section 5.3, 6 over multiple dataset even on smaller datasets like CIFAR-10/100, STL10 etc.

2. Some citation may be added:

a. Introduction: Representation learning, ...acquiring condensed [cite] ...

b. However, unlike supervised learning, .... driven empirically: Not true. There has been a host of theoretical work on self-supervised learning, some of them has been mentioned in the related work section and some in the weakness section above.

c. In Related work section: Wang & Isola (2020) ... in an asymptotic setting. Not clear what has been conveyed. Additionally, in next paragraph: There has been attemps ..."in different languages" may be rephrased.

**Strengths And Weaknesses:**

Strengths:

1. Paper is easy to read and understand.
2. The paper attempts to draw a relation between self-supervised loss, InfoNCE, and supervised contrastive loss.


Weaknesses:

1. Overstatement of the scope of the work:
While the title, and narration of the paper is motivated as bridging the gap between supervised and self-supervised contrastive learning, the work is limited to a theoretical formulation of the contrastive InfoNCE/ NT-Xent loss in SimCLR. There are other SSL methods that do not require negative samples like SwAV, SimSiam, Barlow Twins, Self-classifier [A], etc, the paper does not discuss how the proposed formulation can be extended to those methods. The paper also lacks to establish the relation between generative self-supervised methods like Masked-image-modelling, SimMIM, BEIT, etc.
While the authors briefly discuss non-contrastive methods in Section 7, the discussion is cursory and lacks a clear theoretical connection or justification in relation to the proposed formulation.

2. Triplet loss, hard-negative sampling, and role of temperature parameter:
- On page 4, para 2: ... we only repel the prototype representation with maximum similarity. This assumption is quite strong. In InfoNCE loss, the temperature parameter controls the smothness\peakiness of this sampling across the negative samples [Chen et al. (2020a)].
- Empirical study, section 6 again discusses this role of temperature parameter. The \alpha parameter in the proposed formulation (inverse of temperature: \tau, as discussed in section 5.2) has been stated as offering insights into the role of negative sampling and temperature parameters in InfoNCE-type losses. However, there is a host of literate already establishing this connection including Chen et al. (2020a), Wang and Liu (2021), [B, C, D]. Hence, no new insights are provided in this regard.

3. Insufficient empirical analysis for parameter \alpha and \lambda:
-  Section 6, Empirical Study, discusses the balancing factor \alpha and \lambda and provides a balanced contrastive loss that produces the highest accuracy. However, the analysis has only been done on the Imagenet dataset and the empirically derived values of the parameters may not be generalizable to other datasets. The paper should have provided a more comprehensive analysis of the parameter sensitivity across different datasets and tasks. For example, the paper could have included experiments on CIFAR-10 or other datasets to demonstrate the robustness of the proposed method. The paper doesn't provide any discussion on the heuristics of how to select these parameters. The paper should have provided a more comprehensive analysis of the parameter sensitivity across different datasets and different dataset imabalnce-factors.

4. Unclear message of section 5.5 Dataset: balanced.
- The paper states existing literature states the SSL model performs better in a balanced dataset setting in comparison to an imablanced dataset setting, and the proposed formulation also doesn't perform well in an imbalanced dataset setting. However, it is unclear what is the insight from this discussion.

5. Unclear discussion on the role of Similarity measure and normalization:
- The paper states many formulations use cosine similarity and l2 normalization, it is crucial for the success of contrastive learning. While dot product without normalization and Euclidean without l2 normation don't work as well as cosine with normalization.
The paper lacks a justification of why this is the case. Furthermore, there is a better explanation provided for the case of l2 normalization of features in [E]. L2 normalization is also crucial for projecting the representation vector on the unit-hypersphere (Caron et al., 2020;). Overall, there are no new insights from this paragraph.

6. Overstatement of the insights provided by the proposed formulation on Siamese architecture and the notion of asymmetry:
- In section 5.1, the paper states the proposed formulation provides insights into why there exists asymmetry in Siamese architecture. However, it doesn't explain how the formulation explains the architectural asymmetry for example in the case of SimSiam, or BYOL, or symmetry in the case of Barlow twins.


Overall, while the premise of the paper is interesting, the paper lacks sufficient theoretical and empirical justification to support its claims. The paper also fails to establish a clear connection between the proposed formulation and other self-supervised methods, particularly those that do not rely on negative samples. The paper would benefit from a more comprehensive analysis of the proposed method's performance across different datasets and scenarios like dataset imbalance, as well as a clearer discussion of the implications of the findings.


[A] Amrani, E., Karlinsky, L., & Bronstein, A. (2022, October). Self-supervised classification network. In European Conference on Computer Vision (pp. 116-132). Cham: Springer Nature Switzerland.

[B] Kalantidis, Y., Sariyildiz, M. B., Pion, N., Weinzaepfel, P., & Larlus, D. (2020). Hard negative mixing for contrastive learning. Advances in neural information processing systems, 33, 21798-21809.

[C] Zhang, C., Zhang, K., Pham, T. X., Niu, A., Qiao, Z., Yoo, C. D., & Kweon, I. S. (2022). Dual temperature helps contrastive learning without many negative samples: Towards understanding and simplifying moco. In Proceedings of the IEEE/CVF conference on computer vision and pattern recognition (pp. 14441-14450).

[D] Jiang, R., Nguyen, T., Ishwar, P., & Aeron, S. (2024, June). Supervised contrastive learning with hard negative samples. In 2024 International Joint Conference on Neural Networks (IJCNN) (pp. 1-8). IEEE.

[E] Zhang, C., Zhang, K., Zhang, C., Pham, T. X., Yoo, C. D., & Kweon, I. S. (2022). How does simsiam avoid collapse without negative samples? a unified understanding with self-supervised contrastive learning. arXiv preprint arXiv:2203.16262.

---

> ### Author Response · Authors · 2025-04-20
> **Response to Reviewer eQZ2's Comments**
>
> We sincerely thank you for your thoughtful and constructive feedback. As self-supervised learning is a vast and evolving field, our work focuses on a specific aspect of it. In response to your suggestions, we have revised the manuscript (including the title and abstract) to clarify that our focus is on contrastive learning. Your insights have significantly helped us improve the quality and clarity of our work. Please find our detailed responses below.
>
> ---
> ### **[W1] Scope of the work**
> While our formulation and analysis may offer some theoretical insights into non-contrastive methods (especially the attracting component), we acknowledge that your point is valid. Since our work focuses on contrastive learning, we have revised the manuscript to better reflect this scope.
>
>
> ---
> ### **[W2] Temperature parameter**
> As acknowledged, the connection between the temperature parameter and the sharpness of the softmax distribution has been discussed. However, our contribution lies in providing a formal theoretical connection from a principled framework that approximates supervised learning. Specifically, we start from a triplet loss with pseudo-labels (based on hard negative mining) and rigorously show how an InfoNCE-type loss with samples (relaxed using the temperature parameter) arises as an upper bound. This bridges a gap in prior work, where such a connection has not been theoretically grounded.
>
> ---
> ### **[W3] Empirical results on multiple datasets**
>
> We would like to clarify that our empirical analysis is not limited to ImageNet. We also included experiments on CIFAR-10 (see Section A.4.3 and Figure 5 in the Appendix). To demonstrate the importance of selecting appropriate balancing parameters across datasets, we conducted an extensive grid search over $\alpha$ and $\lambda$.
>
> ---
> ### **[W4] Discussion on balanced datasets**
> Thank you for your feedback. The intention of Section 5.5 was to highlight that our theoretical assumption regarding balanced datasets also has a significant impact on practical performance. We have revised the manuscript to make the intention more explicit.
>
>
> ---
> ### **[W5] Discussion on similarity measures**
> Our goal is to show that this practice is consistent with our theoretical framework. By illustrating the role this assumption plays in our derivation, we aim to better position the practice within a theoretical context.
>
> In addition, to our knowledge, this aspect has not been explicitly ablated in prior work. For instance, in Table 5 of the SimCLR paper, multiple components (normalization and temperature) are modified simultaneously, making it difficult to isolate the effect of normalization alone. Therefore, we provide dedicated ablation results to support its importance.
>
> ---
> ### **[W6] Discussion on symmetry and asymmetry**
>
> In Section 5.1, our intention was to suggest that our formulation offers a possible theoretical explanation for the coexistence of both symmetry and asymmetry in the self-supervised learning literature. Averaging over augmentations in our formulation results in lower-variance targets, consistent with the design choice in SimSiam and BYOL to stabilize one encoder. We do not claim to fully explain all architectural choices in self-supervised learning. Rather, we offer a plausible explanation consistent with empirical trends, and we have carefully qualified our statements in Section 5.1.

---

### Decision · Action_Editor_QLPX · 2025-05-18

**Recommendation:** Reject

**Comment:**

While the paper attempts to investigate how supervised contrastive learning and self-supervised contrastive learning are interrelated, reviewers raised a concern about the over-claimed conclusion. AE saw the authors tried to avoid this issue in the revision, but it still persists in the latest version.

One of the most significant points is that this paper indeed does not bridge supervised and self-supervised learning *in theory*, but only experimentally via the prototype representation bias. This analysis approach looks good to me, but the abstract and introduction give an impression that self-supervised learning is theoretically connected to supervised representation learning. Taking an example, AE does not feel like calling this “a mathematical foundation for commonly used contrastive
losses” (in introduction). Note that I do not mean the analysis does not make sense, to make sure.

The tightness of the upper bounds on the attracting and repelling components is not clear. Adding a numerical demonstration to see the tightness of these bounds should enhance the contributions of this paper (it is totally fine to simulate under the control setup where the assumptions are satisfied).
Together with this, both upper bounds are derived from Jensen’s inequality, so it is vital to discuss how likely the equality condition is.

The explanation on the advantage of the asymmetric network branches is a little bit nuanced. The presented logic is like this: Wang et al. (2022) argues the asymmetric variance between the two network branches contributes to better performance; and this paper successfully recovers the asymmetric attracting term in Equation (8) *by assuming* we are interested in supervised representation learning of type Equation (3). However, this does not corroborate the benefit of the asymmetric network but rather remain to provide its reinterpretation.

Additionally, “supervised learning problem” occasionally used in abstract and introduction is ambiguous so that it may sound like supervised classification, but actually this turns out to be supervised representation learning (or metric learning) afterwards. This ambiguity should be avoided to prevent misleading.

Therefore, we encourage the authors to work on polishing the presentation for one more time. Let’s carefully consider what are/aren’t implied by theoretical/experimental results.

**Audience:**

Yes

**Claims And Evidence:**

No

**Resubmission Of Major Revision:**

The authors may consider submitting a major revision at a later time.